# Early-Adulthood Weight Change and Later Physical Activity in Relation to Cardiovascular and All-Cause Mortality: NHANES 1999–2014

**DOI:** 10.3390/nu14234974

**Published:** 2022-11-23

**Authors:** Xinyu Xiao, Chengyao Tang, Xiaobing Zhai, Shiyang Li, Wenzhi Ma, Keyang Liu, Shirai Kokoro, Haytham A. Sheerah, Huiping Zhu, Jinhong Cao

**Affiliations:** 1School of Public Health, Wuhan University, Wuhan 430071, China; 2Biomedical Statistics, Department of Integrated Medicine, Graduate School of Medicine, Osaka University, 2-2 Yamadaoka, Suita-shi 565-0871, Osaka, Japan; 3Center for Artificial Intelligence Driven Drug Discovery, Faculty of Applied Sciences, Macao Polytechnic University, Macau SAR, China; 4Public Health, Department of Social Medicine, Graduate School of Medicine, Osaka University, 2-2 Yamadaoka, Suita-shi 565-0871, Osaka, Japan; 5Ministry of Health, Riyadh 11176, Saudi Arabia; 6Department of Epidemiology and Health Statistics, School of Public Health, Capital Medical University, Beijing 100069, China

**Keywords:** early-adulthood weight change, physical activity, cardiovascular disease, mortality, National Health and Nutrition Examination Survey (NHANES)

## Abstract

Limited evidence investigated the combined influence of early-adulthood weight change and later physical activity on the risk of cardiovascular (CVD) and all-cause mortality. The aim of this study is to explore the associations of early-adulthood weight change and later physical activity with CVD and all-cause mortality. This is a cohort study of 23,193 US adults aged 40 to 85 years from the National Health and Nutrition Examination Survey (NHANES) 1999 to 2014. Cox proportional hazards regression was used to calculate hazard ratios (HRs) with 95% confidence intervals (CIs) of CVD and all-cause mortality associated with early-adulthood weight change and later physical activity. During a median follow-up of 9.2 years, there were 533 and 2734 cases of CVD and all-cause deaths. Compared with being physically inactive, the HRs of the CVD mortality of being physically active were 0.44 (0.26 to 0.73), 0.58 (0.19 to 1.82), 0.38 (0.17 to 0.86) and 0.46 (0.21 to 1.02) among individuals with stable normal, stable obese, non-obese to obese and maximum overweight early-adulthood weight change patterns. Using stable normal patterns that were physically active later as the reference, other early-adulthood weight change patterns did not show a significantly higher risk of CVD mortality when participants were physically active in later life; later physically inactive participants had a significantly increased risk of CVD mortality, with HRs of 2.17 (1.30 to 3.63), 5.32 (2.51 to 11.28), 2.59 (1.29 to 5.18) and 2.63 (1.32 to 5.26) in the stable normal, stable obese, non-obese to obese and maximum overweight groups, respectively. Similar results can be seen in the analyses for all-cause mortality. Our findings suggest that inadequate physical activity worsens the negative impact of unhealthy early-adulthood weight change patterns, which is worthy of being noted in the improvement of public health.

## 1. Introduction

Obesity has been a major concern of public health for a long time. It is linked to numerous adverse health effects and is considered a key risk factor of cardiovascular diseases (CVD), such as hypertension [1], stroke [2] and coronary heart disease [3]. In recent years, more studies paid attention to the weight change of a certain life period rather than the weight of a single time-point. It is pointed out that people who gained weight during adulthood showed an increased risk of stroke [4], CHD [5], CVD [6] and all-cause mortality [7,8]. In addition, weight gain was generally more rapid during young adulthood, and obesity most probably occurs in this period, while weight began to stabilize or even decrease in older adulthood [9,10].

As an instrumental practice of cardiovascular prevention guidelines, the benefits of physical activity with respect to improving cardiovascular health are well recognized. Physical activity is known to have a beneficial influence on many health outcomes among individuals with obesity. Previous studies have shown that increased levels of physical activity reduce CVD and all-cause mortality risks [11,12]. People who perform physical activity above the recommended level according to the current public health guidelines were associated with lower morbidity and mortality [13]. A high level of physical activity contributed to weight loss, which improved cardio-metabolic risk factors, such as the lipid profile [14], *C*-reactive protein [15,16], insulin sensitivity [17,18] and resting blood pressure [19]. Moreover, physical activity showed cellular and systemic physiologic benefits for obese individuals, independent of weight change [20]. However, the effect of physical activity among individuals with different weight change experiences is less clear. Few studies focused on past weight change experience and later physical activity to examine whether later physical activity can offset the detrimental influence of unhealthy weight change in early adulthood, as well as the combined association of these two factors on health.

This cohort study was designed to investigate the effect of later physical activity on CVD and all-cause mortality among people with different weight change experiences in early adulthood, as well as the joint effect of early-adulthood weight change and later physical activity on these health outcomes. We hypothesized that achieving the recommended level of physical activity would reduce CVD and all-cause mortality risks in different groups of early-adulthood weight change. Moreover, inadequate physical activity would worsen the effect of unhealthy early-adulthood weight changes on the primary outcomes.

## 2. Materials and Methods

### 2.1. Study Population

The National Health and Nutrition Examination Survey (NHANES) was conducted by the National Center for Health Statistics (NCHS) of the Centers for Disease Control and Prevention (CDC). The nationally representative data of the noninstitutionalized civilian population in the United States were collected through a multi-stage, stratified, closeted probability sampling method. More details of the sampling method and data collection have been published on the web (www.cdc.gov/nchs/nhanes/index.htm (accessed on 17 July 2022)). Written informed consent was obtained from all participants.

In this analysis, participants aged 40 years or over with follow-up information on mortality and underlying cause of death from the NHANES 1999–2014 datasets were included (*n* = 35,415). Further, we excluded those who were pregnant, those who had been diagnosed with CVD or cancer disease at baseline, those without baseline information on physical activity and those who had missing data on baseline height and weight as well as weight at 25 years old and at 10 years before baseline. Additionally, we excluded those who were obese at the age of 25 but not at 10 years before baseline, defined as obese to non-obese in early adulthood in our study, owing to the relatively small number (*n* = 233) and few CVD death cases (*n* = 7). Finally, a total of 23,193 subjects comprised our study population for our analysis (Figure 1).

### 2.2. Outcome Ascertainment

The NHANES uploaded mortality data documents online. Mortality status was ascertained by probabilistic matching to the National Death Index through 31 December 2015. Details of the matching method are available from the NCHS (www.cdc.gov/nchs/data-linkage/mortality-public.htm (accessed on 17 July 2022)). Underlying causes of deaths were determined according to the codes of ICD-10 (international statistical classification of diseases, 10th revision). The primary outcomes of this cohort study were mortality due to CVD and all causes. We defined deaths from CVD as deaths from either heart diseases (codes I00–I09, I11, I13, I20–I51) or cerebrovascular diseases (codes I60–I69).

### 2.3. Assessments of Early-Adulthood Weight Change

The NHANES body measure examination was conducted by the mobile examination center (MEC), where the baseline weight and height data of participants were collected. Information on weight at 25 years old and at 10 years before baseline was recalled at the baseline surveys. Body mass index (BMI) was calculated as weight (kg) divided by the square of height (m^2^). We calculated BMI at 25 years old, at 10 years before baseline as well as at baseline, and we classified BMI at each time point as underweight and normal weight (<25), overweight (25.0–29.9) and obese (≥30.0). Using BMI at 25 years old and at 10 years before baseline, early-adulthood weight change can be categorized into five patterns: stable normal pattern (<25.0 at both time points), maximum overweight pattern (25.0–29.9 at either time point), obese to non-obese pattern (≥30.0 at 25 years old and <30.0 at 10 years before baseline), non-obese to obese pattern (<30.0 at 25 years old and ≥30.0 at 10 years before baseline) and stable obesity (≥30.0 at both time points) [21]. As mentioned before, we excluded participants with obese to non-obese patterns in early adulthood.

### 2.4. Assessment of Later Physical Activity

In our study, physical activity at baseline was considered as the later physical activity level. It was assessed by an interview questionnaire and composed of physical activity at work/domestic, in transport/travel and in leisure time. For the period 1999–2004, each of the activities was awarded an energy expenditure on metabolic equivalents (METs) according to the standard designed by Ainsworth [22]. We multiplied the standard MET value of each activity by the minutes spent on each activity in a typical week to obtain the MET min per week of each activity, of which the total MET min per week for physical activity was the sum. The assessment method was consistent with previous studies [23,24]. For the period 2005–2014, physical activity was assessed according to the Global Physical Activity Questionnaire (GPAQ) created by The World Health Organization (WHO), in which the intensity (moderate or vigorous) of physical activity was also taken into consideration. The total MET min per week for physical activity can be obtained following the GPAQ protocol.

The WHO suggested that adults should engage in physical activity for at least 75 min at a vigorous intensity, 150 min at a moderate intensity or an equivalent combination of moderate- and vigorous-intensity activity, achieving at least 600 METs per minute in a typical week [25]. Thus, we defined the physically inactive group as those who did not meet the recommended threshold of physical activity (<600 MET min/week), and the physically active group was defined as those who achieved it (≥600 MET min/week).

### 2.5. Covariate Assessment

Information about participants’ age, sex, race/ethnicity, education, ratio of family income to poverty, total cholesterol, high-density lipoprotein (HDL), alcohol drinking status, smoking status, hypertension and diabetes was available through baseline questionnaires or measurements. Dietary data were obtained by 24-h dietary recall interviews, including the types and amounts of all foods and beverages consumed during the 24 h period prior to the interview (midnight to midnight). NHANES estimated the intakes of energy, nutrients and other food components from those foods and beverages with the USDA’s Food and Nutrient Database for Dietary Studies (FNDDS). The FNDDS includes comprehensive information that can be used to code individual foods and portion sizes reported by participants and also includes nutrient values for calculating nutrient intakes. The estimations of carbohydrate intake, dietary fiber, protein intake, fat intake and energy intake were used in our study. In detail, race/ethnicity was categorized into Mexican-American, non-Hispanic white, non-Hispanic black or other. Education level was classified as under high school, high school and above high school. Based on the participants’ responses to questions about whether they have smoked at least 100 cigarettes until now and whether they were smoking at present, we divided them into nonsmoker, past smoker and current smoker. Alcohol intake status was grouped as none (0 g/day), moderate drinking (0.1 to 27.9 g/day for men and 0.1 to 13.9 g/day for women) and heavy drinking (≥28 g/day for men and ≥14 g/d for women). Hypertension was determined according to the question “Have you ever been told by a doctor or other health professional that you had hypertension, also called high blood pressure”. Participants with at least one of the following conditions were defined as having diabetes: (1) self-reported doctor diagnosis of diabetes; (2) self-reported use of insulin or oral hypoglycemic medication; (3) fasting glucose ≥ 7.0 mmol/L; (4) glycated hemoglobin A1c (HbA1c) ≥ 6.5%.

### 2.6. Statistical Analysis

Our statistical analyses utilized appropriate sample weights and strata corresponding to the complex, multistage, stratified and cluster-sampling design of NHANES. We classified participants into eight groups according to four early-adulthood weight change patterns (stable normal, stable obesity, non-obese to obese and maximum overweight) and two later physical activity levels (physically inactive and physically active), across which baseline characteristics were described.

Person-years of follow-up were calculated from the baseline (baseline year) to their first endpoint in this follow-up as follows: death, moving out or the end of follow-up, whichever came first. The proportional hazard assumption was checked and was not violated in our study. After that, we first used Cox proportional hazard regression models to estimate the hazard ratios (HRs) and 95% confidence intervals (CIs) of CVD and all-cause mortality risks associated with early-adulthood weight change and later physical activity with the following covariates: age, sex and race/ethnicity in model 1; model 1 plus education, family income, baseline BMI, total cholesterol, HDL cholesterol, carbohydrate intake, dietary fiber, protein intake, fat intake, energy intake, alcohol drinking status, smoking status, hypertension, diabetes and later physical activity (only in the analysis for early-adulthood weight change) or early-adulthood weight change pattern (only in the analysis for later physical activity) in model 2. In the analysis with early-adulthood weight change pattern as the exposure, HRs were calculated with a stable normal pattern as the reference group, and in the analysis with later physical activity as the exposure, HRs were calculated with the physically inactive level as the reference group. After that, the associations between early-adulthood weight change pattern and mortality in subgroups of age, sex, baseline BMI and physical activity were investigated. We further conducted a stratified analysis by early-adulthood weight change to examine the associations of later physical activity with CVD and all-cause mortality risks. To assess the combined effects of early-adulthood weight change and later physical activity, we investigated the HRs and 95% CIs of CVD and all-cause mortality in eight groups, as classified before, using the group of the stable normal pattern together with the physically active level as the reference. All statistical analyses were conducted using survey modules of SAS software version 9.4 (SAS Institute, Cary, North Carolina). Two-sided *p* values <0.05 were considered statistically significant in our study.

## 3. Results

### 3.1. Population Characteristics

Since no interaction with sex was found for the association of early-adulthood weight change and later physical activity, we pooled the results of men and women to present in the main analysis. The baseline characteristics of the eight groups categorized by early-adulthood weight change patterns and later physical activity level are shown in Table 1. Among 23,193 participants (mean age 53.6 years, 49.3% men), 42.7% (*n* = 9088) were in the stable normal category, 5.5% (*n* = 1297) were in the stable obesity category, 15.1% (*n* = 3880) were in the non-obese to obese category and 36.7% (*n* = 8928) were in the maximum overweight category. Moreover, 47.9% (*n* = 11,764) were physically inactive, and 52.1% (*n* = 11,429) were physically active. Compared with participants who kept a stable normal weight in early adulthood, those with other weight change experiences tended to be men, non-Hispanic black or Mexican-American, less educated and poorer and have hypertension, diabetes and more energy and fat intakes. Participants in the physically inactive group were composed of more women and non-Hispanic white people, were more likely to have diabetes and hypertension, and took in less carbohydrates, dietary fiber, energy and fat. In addition, individuals with a stable normal weight change pattern tended to be physically active, while those in other groups tended to be physically inactive.

### 3.2. Independent Association of Early-Adulthood Weight Change Pattern and Later Physical Activity with CVD and All-Cause Mortality

During 206,401 person-years of follow-up (median follow-up 9.2 years), 2734 participants died, and 533 of them died from CVD. Table 2 shows the association between early-adulthood weight change, later physical activity and mortality risks, using the stable normal group and the physically inactive group as the reference, respectively. Expectedly, after adjustment for age, sex and race/ethnicity, participants who were stable obese in early adulthood had increased risks of CVD and all-cause mortality, with respective HRs of 3.75 (1.81,7.76) and 1.89 (1.41,2.54). In the fully adjusted model, the HRs for CVD and all-cause mortality for stable obese participants were 2.51 (1.41,4.47) and 1.69 (1.22,2.35). As for the non-obese to obese group, the HRs were 1.20 (0.66,2.16) for CVD mortality and 1.11 (0.90,1.38) for all-cause mortality. As for the maximum overweight group, the HRs of CVD and all-cause mortality were 1.30 (0.88,1.93) and 1.14 (0.92,1.40). Being physically active was associated with significantly lower risks for both CVD and all-cause mortality than being physically inactive; the HRs were 0.51 (0.33,0.81) for CVD mortality and 0.67 (0.55,0.81) for all-cause mortality.

### 3.3. Associations of Early-Adulthood Weight Change Pattern with CVD and All-Cause Mortality among Different Subgroups

Figure 2 shows the associations between early-adulthood weight change pattern and CVD, all-cause mortality stratified by age, gender, baseline BMI and later physical activity. Compared with the stable normal pattern, all three unhealthy weight change experiences had negative effects on CVD mortality in the subgroups of age ≥60 years old and baseline BMI >25 kg/m^2^, while no such results were found in the subgroups of age <60 years old and baseline BMI ≤25 kg/m^2^. Moreover, the stable obese pattern was associated with a significantly higher risk of CVD mortality among female and physically inactive participants only, with HRs of 5.30 (2.33 to 12.09) and 2.62 (1.43 to 4.81), respectively. The results of the analysis for all-cause mortality were mostly similar. A relation between a stable obese pattern and a higher all-cause mortality was found in both the male and female subgroups, but it was stronger in females.

### 3.4. Interaction and Joint Analysis of Early-Adulthood Weight Change Pattern and Later Physical Activity with CVD and All-Cause Mortality

No significant interaction was observed between early-adulthood weight change and later physical activity on both CVD and all-cause mortality. As shown in Figure 3, being physically active was associated with a lower risk of CVD mortality among individuals of the stable normal group and non-obese to obese group, with HRs of 0.44 (0.26 to 0.73) and 0.38 (0.17 to 0.86), respectively, whereas the HR of the physically active was 0.58 (0.19 to 1.82) in the stable obese group. In the analysis for all-cause mortality, the HRs for those who were physically active compared with those who were physically inactive were 0.73 (0.58 to 0.91) among individuals of the stable normal group, 0.46 (0.22 to 0.93) among those in the stable obese group, 0.75 (0.53 to 1.08) among those in the non-obese to obese group and 0.58 (0.43 to 0.80) among those in the maximum overweight group. Being physically active led to a significantly lower risk of all-cause mortality in almost all early-adulthood weight change patterns, except for the non-obese to obese group.

Figure 4 shows the combined effects of early-adulthood weight change and later physical activity on the primary outcomes. Using the stable normal pattern and being physically active as the reference, other weight change patterns did not show a significantly higher risk of CVD mortality when participants were physically active in later life. However, later physically inactive participants with all early-adulthood weight change patterns had a significantly higher risk of CVD mortality, with hazard ratios of 2.17 (1.30 to 3.63), 5.32 (2.51 to 11.28), 2.59 (1.29 to 5.18) and 2.63 (1.32 to 5.26) in the stable normal group, stable obese group, non-obese to obese group and maximum overweight group, respectively. Similar results can be seen in the analysis for all-cause mortality. There were significantly higher risks of all-cause mortality in groups with a physically inactive level but not in groups with a physically active level.

## 4. Discussion

In this large cohort study of the nationally representative US population, being stable obese across early adulthood was associated with higher risks of CVD and all-cause mortality. Being physically active in later life reduced the risk of CVD mortality among individuals with stable normal or non-obese to obese weight change patterns in early adulthood; it also reduced the risk of all-cause mortality among individuals with stable normal, stable obese or maximum overweight weight change patterns in early adulthood. In the analysis of joint effect, individuals with unhealthy weight change patterns in early adulthood combined with being physically inactive in later life showed significantly higher risks of CVD and all-cause mortality. However, individuals with unhealthy weight change patterns in early adulthood combined with being physically active in later life did not show higher risks of these primary outcomes. The results were independent of demographic factors, socioeconomic factors, dietary factors, lifestyle factors as well as common risk factors of CVD.

The relationship between body weight and mortality has been widely discussed in previous cohort studies [26,27,28]. A growing number of studies divert their attention to weight variation and health considering that weight changes often happen in adulthood. A prospective cohort study of 1657 Finnish men showed that the largest weight gain from 25 years old to midlife can be a predictor of long-term death, with a relative HR of 1.39 (95% CI: 1.12–1.73) [29]. Utilizing the data of participants recruited in the National Institutes of Health-AARP Diet and Health Study, Adams et al. reported that weight gain in young (18–35 years) and middle (35–50 years) adulthood was strongly associated with increased mortality, with 66% and 61% higher risks compared to stable weight, respectively [30]. Additionally, a cohort study based on 36,051 people from NHANES found that being stable obese from young to middle adulthood was associated with increased risks of all-cause (HR: 1.72; 95% CI: 1.52–1.95) and heart disease mortality (HR: 2.57; 95% CI: 1.92–3.43) [21]. As is known to all, physical activity has beneficial effects on health outcomes. Some explored the combined effect of weight and physical activity on mortality risk. A study of 185,412 US participants indicated that a high physical activity compensated the elevated mortality risk linked with obesity to some degree [31]. Another cohort study including 22,476 US adults concluded that overweight and obese individuals benefit from any level of physical activity, which lowered their 10-year CVD risk [32]. However, there is a limitation in these studies: body weight information was collected at just a single time point. To the best of our knowledge, our study is the first analysis exploring early-adulthood weight change and later physical activity in relation to CVD and all-cause mortality.

In our analysis, individuals with stable obesity during the span of early adulthood showed higher risks of CVD and all-cause mortality, whereas there is a null association in the non-obese to obese group and the maximum overweight group. Weight changes are common across adulthood. Body composition changes with increasing age may explain this. As fat takes the place of lean mass, it is necessary for the body to gain additional weight to keep the same lean mass [30]. According to our study, long-standing obesity should be avoided to reduce the risks of mortality.

Our study also revealed that the detrimental influence of unhealthy weight change experiences, especially the stable obese pattern, was more significant in elderly people above 60 years old, females, overweight and obese people and later physically inactive people, suggesting that individuals with these characteristics need to pay more attention to the possible higher CVD and all-cause mortality risk brought by unhealthy weight changes in the past. Notably, unlike our analysis, other studies mostly examined the associations of initial BMI and subsequent weight change. A cohort study of 13,104 Americans aged 50 years and over reported that weight gain was related with excess mortality only among people with an initial BMI over 35 [33]. Another study including 331,900 Korean adults aged ≥40 years found that weight gain in obese adults was associated with a higher incidence of disability [34]. According to our study, baseline BMI (collected after early-adulthood weight change, in our study) seems to be a modifier between past weight change experiences and healthy outcomes. If people who experienced unhealthy weight changes before still kept an unhealthy BMI later, they are susceptible to past unhealthy weight changes. It also underlined the importance of adjusting for baseline BMI in our analysis of the relation between early-adulthood weight change, later physical activity and health outcomes. In addition to the results of baseline BMI subgroups, the obviously higher risk of CVD and all-cause mortality due to stable obese pattern in early-adulthood among later physically inactive participants highlighted that the effect of weight change experience may be affected by later physical activity status.

Consistent with previous studies [11,12,13], being physically active led to lower mortality risks, obviously. When we examined the associations between later physical activity and mortality in different early-adulthood weight change patterns, we found that a high level of later physical activity was associated with lower risks of CVD and all-cause mortality under most circumstances, suggesting that adequate physical activity in later life is worthy regardless of weight change experience, as a bonus for health.

Furthermore, our study found that, compared with physically active participants who kept a stable normal weight in early adulthood, those who were physically active and had an unhealthy weight change experience in early adulthood did not show significantly higher mortality risks, while those who were physically inactive and had an unhealthy weight change experience in early adulthood did. It is suggested that the detrimental impact of unhealthy weight changes in early adulthood came out when combined with inadequate physical activity in later life. According to prior studies, obesity contributes to a range of metabolic disorders and disease processes through pathophysiologic changes in multiple organ systems. It affects the cardiovascular system directly. Hemodynamic changes are closely related with it, and it also leads to variations in the cardiac and endothelial structure and function [35], especially in individuals with long-term obesity [36]. Additionally, adipose tissue is a complex organ synthesizing pro-inflammatory adipokines such as tumor necrosis factor-α, interleukin-6, interleukin-1β and resistin [37]. Elevated levels of pro-inflammatory adipokines lead to chronic inflammation and further exacerbate CVD and metabolic disorders [38]. However, the negative effect induced in early adulthood is not fixed. Cardiometabolic risk biomarkers linked to obesity can be changed with physical activity [39]. A study of young European adults concluded that the highest level of physical activity was associated with a lower fat mass, leptin and interleukin 6, and some of the detrimental influence linked with overweightness and obesity was diminished due to physical activity [40]. Physical activity intervention programs targeted at overweight and obese adolescents were also reported to improve body composition [41,42]. These findings may partly explain why the combination of an unhealthy weight change experience and later physical inactivity showed significantly higher risks for CVD and all-cause mortality, whereas an unhealthy weight change experience seems not to matter if one is physically active later.

Our findings highlight the necessity of achieving the recommended level of physical activity for people with unhealthy weight change patterns in early adulthood. It is more urgent to be physically active from now on than to regret not keeping a normal weight before.

The strengths of our study include a large cohort based on the nationally representative population, which allowed us to analyze combined groups of early weight change and later physical activity; the prospective design provides sufficient follow-up information and a sufficient number of outcomes to analyze; the comprehensive data collected in NHANES enabled us to control the possible confounding influence caused by plenty of demographic, socioeconomic, lifestyle and dietary factors.

Our study also has several limitations. First, weight data at the age of 25 years and 10 years before baseline were recalled and self-reported. This may lead to misclassification bias. Second, later physical activity may change or fluctuate, but related information was not available, so our findings are limited. Third, both early-adulthood weight and later physical activity may change due to unreported illnesses or adverse life experience, thus affecting the final results.

Further studies need to explore whether weight change patterns at a young age and later physical activity interact with other common health-related factors such as sleep, diet and so on. Genetic factors should be taken into consideration, as recent research indicates that genetic factors play an important role in the influence of excessive body weight [43] and physical activity [44]. Moreover, considering that BMI may not reflect fat mass and lean mass, it is necessary to further explore the possible association of changes in fat mass and lean mass, physical activity and health outcomes, which can expand the results of our study.

## 5. Conclusions

Based on this large nationwide US cohort study, being physically active strongly lowered the risks of CVD and all-cause mortality in almost all early-adulthood weight change groups, which highlights the necessity of a high level of later physical activity, regardless of weight change patterns in early adulthood. In addition, compared with physically active participants who kept a stable normal weight in early adulthood, physically active participants with other early-adulthood weight change patterns did not show a significantly higher risk of primary outcomes, while physically inactive participants with other early-adulthood weight change patterns showed a higher risk of primary outcomes. Our study suggests that a low level of physical activity worsens the negative impact of unhealthy early-adulthood weight change patterns, which can be another important argument for adequate physical activity. According to our findings, it is beneficial to avoid unhealthy early-adulthood weight changes and keep physically active. For those who have already experienced unhealthy weight changes during early adulthood, achieving the recommended level of physical activity should be encouraged, no matter what specific weight change patterns they have. We argue for the public to raise the awareness that later physical activity plays an important role in improving the cardiovascular health of individuals with unhealthy weight changes at a young age.

## Figures and Tables

**Figure 1 nutrients-14-04974-f001:**
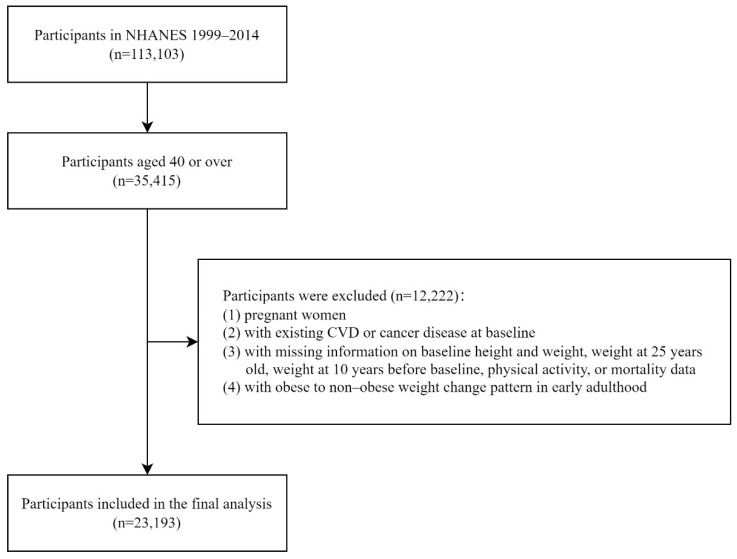
Flowchart of participants included in the analysis.

**Figure 2 nutrients-14-04974-f002:**
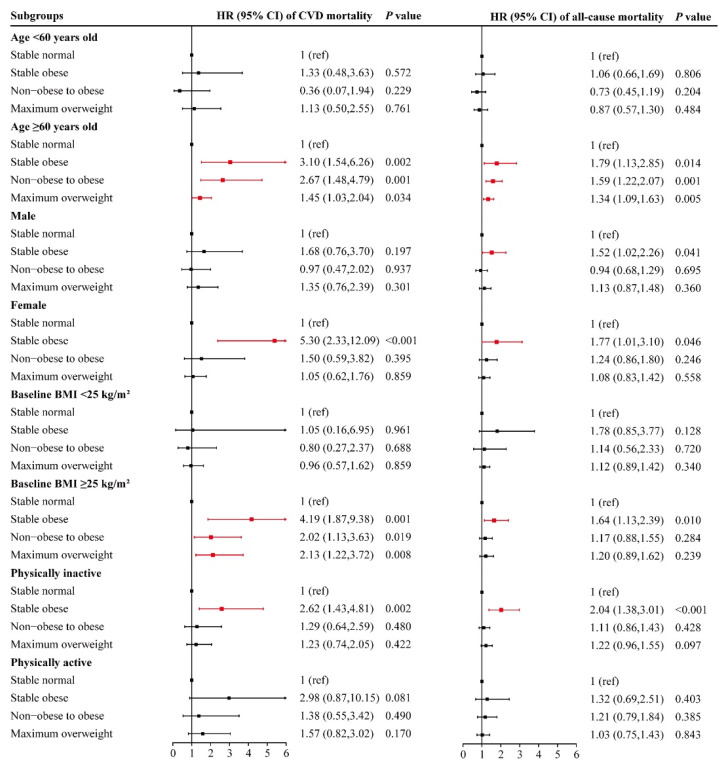
Associations between early-adulthood weight change pattern and CVD, all-cause mortality stratified by age, sex, baseline BMI and physical activity. HRs were adjusted for age (not adjusted in subgroup analysis by age), sex (not adjusted in subgroup analysis by sex), race/ethnicity, education, family income, baseline BMI (not adjusted in subgroup analysis by baseline BMI), physical activity (not adjusted in subgroup analysis by physical activity), total cholesterol, HDL cholesterol, carbohydrate intake, dietary fiber, protein intake, fat intake, energy intake, alcohol drinking status, smoking status, hypertension and diabetes in the analysis for CVD mortality.

**Figure 3 nutrients-14-04974-f003:**
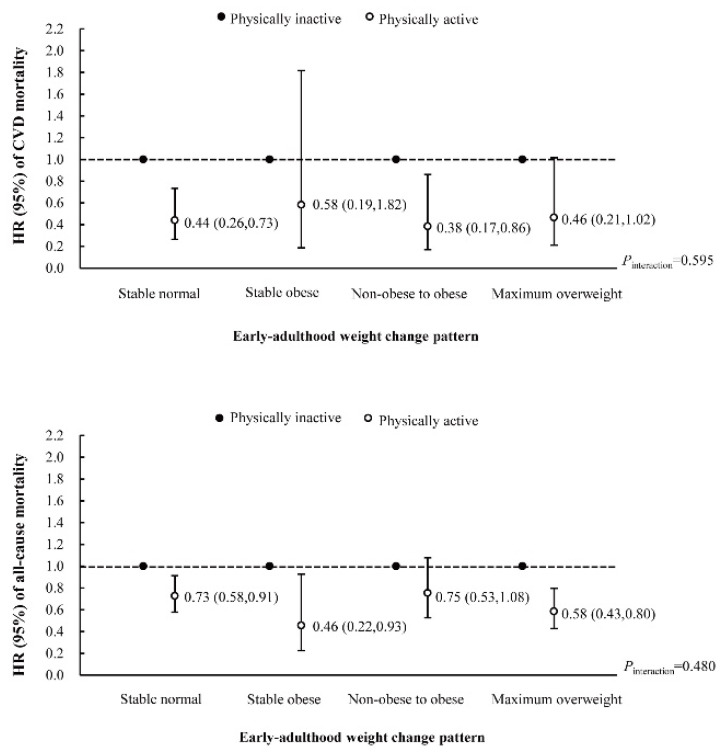
Hazard ratios (95% CIs) of CVD and all-cause mortality according to later physical activity stratified by early-adulthood weight change pattern. Adjusted for age, sex, race/ethnicity, education, family income, baseline BMI, total cholesterol, HDL cholesterol, carbohydrate intake, dietary fiber, protein intake, fat intake, energy intake, alcohol drinking status, smoking status, hypertension and diabetes.

**Figure 4 nutrients-14-04974-f004:**
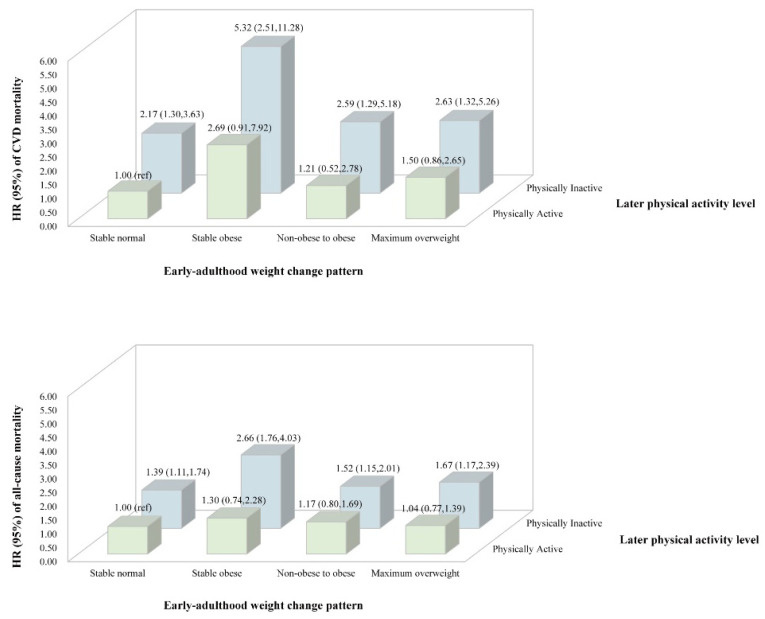
Combined effects of early-adulthood weight change pattern and later physical activity on CVD and all-cause mortality. Hazard ratios (95% CIs) were calculated using Cox proportional hazards regression analysis after adjusting for age, sex, race/ethnicity, education, family income, baseline BMI, total cholesterol, HDL cholesterol, carbohydrate intake, dietary fiber, protein intake, fat intake, energy intake, alcohol drinking status, smoking status, hypertension and diabetes.

**Table 1 nutrients-14-04974-t001:** Baseline characteristics of the study participants, according to early-adulthood weight change pattern and later physical activity.

Characteristic	Stable Normal		Stable Obese		Non-Obese to Obese		Maximum Overweight	
	Inactive	Active	Inactive	Active	Inactive	Active	Inactive	Active
Number of participants	4415	4673	717	580	2119	1761	4513	4415
Age (years), *n* (%)								
<60	2880(77.07)	3294(79.62)	494(81.29)	467(87.15)	1010(62.49)	994(68.86)	2450(68.88)	2704(73.17)
≥60	1535(22.93)	1379(20.38)	223(18.71)	113(12.85)	1109(37.51)	767(31.14)	2063(31.12)	1711(26.83)
Sex, *n* (%)								
Male	1658(35.38)	1930(36.96)	316(45.64)	347(66.41)	905(43.68)	966(56.41)	2493(56.91)	2949(69.00)
Female	2757(64.62)	2743(63.04)	401(54.36)	233(33.59)	1214(56.32)	795(43.59)	2020(43.09)	1466(31.00)
Race/ethnicity, *n* (%)								
Mexican-American	638(4.22)	518(3.48)	117(5.18)	83(5.60)	419(6.33)	352(6.56)	983(6.93)	711(5.14)
Non-Hispanic white	2177(73.67)	2620(78.82)	285(68.89)	277(72.55)	959(74.36)	848(76.12)	2038(72.78)	2320(78.62)
Non-Hispanic black	957(10.29)	787(7.24)	245(17.64)	164(13.35)	543(12.59)	377(9.80)	977(11.14)	817(8.33)
Other	643(11.81)	748(10.46)	70(8.29)	56(8.49)	198(6.73)	184(7.52)	515(9.16)	567(7.91)
Education, *n* (%)								
<High school	1099(14.57)	854(10.79)	205(16.30)	125(12.58)	678(19.06)	402(12.77)	1392(17.99)	949(12.54)
High school	998(22.28)	989(20.35)	183(28.01)	158(29.88)	508(27.64)	428(26.85)	1026(24.67)	966(21.89)
>High school	2313(63.15)	2827(68.86)	327(55.68)	297(57.54)	930(53.30)	931(60.38)	2092(57.34)	2494(65.57)
Ratio of family income to poverty, *n* (%)								
≤1.30	965(14.43)	961(13.07)	195(20.66)	158(19.13)	528(16.87)	369(12.61)	989(14.39)	848(11.81)
1.31–3.50	1432(31.61)	1371(27.38)	247(34.86)	195(36.06)	727(32.36)	628(36.26)	1505(30.72)	1351(27.80)
>3.50	1659(53.96)	2008(59.55)	205(44.49)	197(44.81)	691(50.77)	660(51.13)	1688(54.89)	1901(60.39)
Baseline BMI (kg/m^2^), mean ± SD	24.95 ± 4.09	24.49 ± 3.65	39.54 ± 8.86	36.75 ± 7.43	34.66 ± 5.96	33.63 ± 5.30	29.28 ± 4.23	29.06 ± 4.02
Total cholesterol (mg/dL), mean ± SD	207.97 ± 40.34	207.72 ± 40.07	195.06 ± 43.04	197.69 ± 41.21	202.93 ± 44.77	203.55 ± 40.90	207.03 ± 41.2	208.39 ± 41.7
HDL cholesterol (mg/dL), mean ± SD	58.31 ± 17.95	59.52 ± 17.55	49.05 ± 13.51	47.58 ± 12.81	50.35 ± 15.11	50.11 ± 13.54	51.42 ± 14.57	52.21 ± 15.76
Carbohydrate intake (gm), median (IQR)	225.7(134.1)	235.6(145.3)	224.6(144.0)	253.3(167.9)	213.7(137.7)	231.8(138.9)	228.8(136.7)	242.6(147.3)
Dietary fiber (gm), median (IQR)	13.8(10.8)	15.5(13.2)	13.6(10.8)	14.7(11.8)	13.7(10.7)	15.3(12.2)	14.5(11.7)	16.0(12.0)
Protein intake (gm), median (IQR)	68.6(44.0)	71.8(40.4)	73.3(36.5)	67.6(54.0)	72.9(50.8)	73.9(54.6)	76.8(45.3)	77.8(48.2)
Fat intake (gm), median (IQR)	65.3(49.2)	69(53.5)	73.4(59.3)	81.9(66.9)	65.8(52.3)	78.9(56.7)	70.2(53.2)	75.5(54.0)
Energy intake (kcal), median (IQR)	1828.5(1050.8)	1911(1079.0)	1938(1186.6)	2187.5(1248.0)	1732.0(1097.0)	2014.0(1155.0)	1918.0(1155.0)	2046.5(1195.5)
Alcohol drinking status, *n* (%)								
Nondrinker	3005(68.49)	2923(61.70)	575(80.78)	422(73.09)	1633(76.63)	1266(71.42)	3118(67.30)	2867(63.40)
Moderate drinking	446(11.40)	569(12.80)	34(7.11)	42(7.89)	176(10.91)	179(10.68)	480(12.19)	552(13.75)
Heavy drinking	737(20.10)	913(25.50)	78(12.11)	90(19.02)	202(12.46)	257(17.90)	699(20.52)	809(22.85)
Smoking status, *n* (%)								
Nonsmoker	2290(52.71)	2350(51.01)	408(55.94)	299(54.49)	1184(56.10)	928(52.43)	2305(50.67)	2307(53.55)
Former smoker	1149(26.79)	1295(28.28)	188(25.59)	143(26.32)	647(29.21)	564(32.16)	1385(30.95)	1412(32.40)
Current smoker	971(20.50)	1026(20.71)	120(18.48)	138(19.20)	287(14.69)	269(15.41)	820(18.37)	695(14.05)
Hypertension, *n* (%)								
Yes	1228(22.83)	1087(21.00)	426(61.08)	294(46.23)	1206(51.36)	874(45.57)	1812(34.81)	1641(33.17)
No	3178(77.17)	3575(79.00)	283(38.92)	285(53.77)	906(48.64)	882(54.43)	2684(65.19)	2768(66.83)
Diabetes, *n* (%)								
Yes	319(4.95)	253(3.58)	262(31.74)	196(29.11)	693(24.81)	449(19.28)	802(12.37)	599(9.54)
No	4096(95.05)	4420(96.42)	455(68.26)	384(70.89)	1426(75.19)	1312(80.72)	3711(87.63)	3816(90.46)

Abbreviations: HDL, high-density lipoprotein; SD, standard deviation; IQR, interquartile range.

**Table 2 nutrients-14-04974-t002:** Hazard ratios (95% CIs) of CVD and all-cause mortality according to early-adulthood weight change pattern and later physical activity.

Characteristic	Early-Adulthood Weight Change Pattern				Later Physical Activity	
	Stable Normal	Stable Obese	Non-Obese to Obese	Maximum Overweight	Inactive	Active
CVD mortality						
No. of participants	9088	1297	3880	8928	11,764	11,429
Deaths/person-years	149/83,876	40/10,157	112/32,028	232/80,340	370/108,840	163/97,561
Unadjusted	1.00	3.58 (1.79,7.16)	2.20 (1.37,3.54)	1.89 (1.30,2.76)	1.00	0.47 (0.32,0.71)
Model 1	1.00	3.75 (1.81,7.76)	1.62 (1.00,2.64)	1.40 (0.94,2.08)	1.00	0.47 (0.31,0.71)
Model 2	1.00	2.51 (1.41,4.47)	1.20 (0.66,2.16)	1.30 (0.88,1.93)	1.00	0.51 (0.33,0.81)
All-cause mortality						
No. of participants	9088	1297	3880	8928	11,764	11,429
Deaths/person-years	938/83,876	142/10,157	482/32,028	1172/80,340	1771/108,840	963/97,561
Unadjusted	1.00	1.68 (1.26,2.26)	1.53 (1.25,1.87)	1.40 (1.17,1.69)	1.00	0.62 (0.52,0.75)
Model 1	1.00	1.89 (1.41,2.54)	1.15 (0.95,1.39)	1.10 (0.90,1.33)	1.00	0.62 (0.52,0.75)
Model 2	1.00	1.69 (1.22,2.35)	1.11 (0.90,1.38)	1.14 (0.92,1.40)	1.00	0.67 (0.55,0.81)

Model 1 was adjusted for age, sex and race/ethnicity; Model 2 was further adjusted for education, family income, baseline BMI, total cholesterol, HDL cholesterol, carbohydrate intake, dietary fiber, protein intake, fat intake, energy intake, alcohol drinking status, smoking status, hypertension and diabetes and mutually adjusted for weight change patterns or physical activity, as appropriate.

## Data Availability

The datasets generated and analyzed during the current study are available in the National Health and Nutrition Examination Survey, https://www.cdc.gov/nchs/nhanes/ (accessed on 17 July 2022).

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
