# Peer review of "Early-Adulthood Weight Change and Later Physical Activity in Relation to Cardiovascular and All-Cause Mortality: NHANES 1999–2014"

_nutrients, 2022, doi:10.3390/nu14234974_

Round 1

Reviewer 1 Report

The aim of this study was to explore the associations of early-adulthood weight change and later physical activity with CVD and all-cause mortality.

The authors concluded that the inadequate physical activity worsens the negative impact of unhealthy early-adulthood weight change patterns, worthy of being noted in the improvement of public health.

The manuscript is well written and present interesting findings. However, some modifications are required:

Based on the rational of the study and the data of previous literature, I suggest that some hypothesis are needed at the end of the Introduction section.

Also, I suggest adding some practical recommendations at the end of the manuscript.

Author Response

Thank you for your comments. The comments are all valuable and helpful for revising and improving our paper. We have taken all these comments into account and made correction accordingly. Please see the attachment.

Reviewer 2 Report

Thjis study is very well written. This study was designed to investigate the effect of later physical activity on CVD and all-cause mortality among people with different weight change experiences in early adulthood. However, I have major revisions:

-Introduction is a good, but what is hypothesis of study?

-Methods: Dietary food intake variables were few explained in the methods. Please, to clarify it.

-Results: What is protein intake?

-Discussion: First paragraph of discussion is very confusing. Please, rewrite.

Limitation of study study is no evaluation of lean body mass, which is associated with mortality and CVD risk. Please, to add a explanation for it.

Strong point is a large sample size which strenghts the data and power sample.

Author Response

(The authors gave the same response as above.)

Round 2

Reviewer 1 Report

I suggest that this version is suitable for publication 

Reviewer 2 Report

I agree with presented from.